# Cell migration guided by long-lived spatial memory

Joseph d'Alessandro [1,4✉], Alex Barbier--Chebbah[2,4], Victor Cellerin[1], Olivier Benichou[2], René Marc Mège [1], Raphaël Voituriez [3✉] & Benoît Ladoux [1✉]

Living cells actively migrate in their environment to perform key biological functions—from unicellular organisms looking for food to single cells such as fibroblasts, leukocytes or cancer cells that can shape, patrol or invade tissues. Cell migration results from complex intracellular processes that enable cell self-propulsion, and has been shown to also integrate various chemical or physical extracellular signals. While it is established that cells can modify their environment by depositing biochemical signals or mechanically remodelling the extracellular matrix, the impact of such self-induced environmental perturbations on cell trajectories at various scales remains unexplored. Here, we show that cells can retrieve their path: by confining motile cells on 1D and 2D micropatterned surfaces, we demonstrate that they leave long-lived physicochemical footprints along their way, which determine their future path. On this basis, we argue that cell trajectories belong to the general class of self-interacting random walks, and show that self-interactions can rule large scale exploration by inducing long-lived ageing, subdiffusion and anomalous first-passage statistics. Altogether, our joint experimental and theoretical approach points to a generic coupling between motile cells and their environment, which endows cells with a spatial memory of their path and can dramatically change their space exploration.

[1] Université de Paris, CNRS, Institut Jacques Monod, Paris F-75006, France. [2] Laboratoire de Physique Théorique de la Matière Condensée, CNRS/Sorbonne Université, Paris, France. [3] Laboratoire Jean Perrin and Laboratoire de Physique Théorique de la Matière Condensée, CNRS/Sorbonne Université, Paris, France. [4] These authors contributed equally. Joseph d'Alessandro, Alex Barbier-Chebbah. ✉email: joseph.dalessandro@ijm.fr; voiturie@lptmc.jussieu.fr; benoit.ladoux@ijm.fr

Cell migration is essential for fundamental phases of development and adult life, including embryogenesis, wound healing, and inflammatory responses[1]; it generically results from the active dynamics of its intracellular components—most prominently the cytoskeleton—which generate propulsion forces and determine the cell front-rear polarity[2,3]. The cytoskeleton spatio-temporal dynamics is controlled by complex regulatory networks[4], and can be characterized by both deterministic and stochastic components[5–8]. The integration over time of this complex intracellular dynamics determines the large scale properties of cell trajectories, which can, in turn, be used as accessible read-outs to infer intracellular properties[9–11], as well as cell interactions with the environment[12–15] or with neighbouring cells[16–18]. In vivo, cells interact with various extra-cellular environments with a broad range of biochemical and biomechanical properties[2]. These interactions have been shown to be two-way: environmental cues directly affect cell shape, migration, and polarity[19–21], and in turn, cells actively contribute to remodel their environment[22,23]. So far, however, both ways have been analysed independently, and the feed-back of cell-induced environmental remodelling on large scale properties of cell migration has remained unexplored.

To overcome the inherent complexity of the analysis of cell migration in 3D in vivo environments, the design of micropatterned surfaces has proven to be a powerful approach[24–26]. In such in vitro set-ups, and especially in 1D settings, the reduced dimensionality of the cellular environment allows for an extensive quantitative analysis of the phase space roamed by migrating cells. In particular, such 1D assays have revealed striking deterministic features in cell motility patterns, while cell paths in higher dimensions remain seemingly random[8,18,22,27,28]. Moreover, many of the cell migration features on a 1D substrate can mimic cell behaviour in 3D matrix[22].

## Results

**Isolated cells exhibit regular oscillations.** To dissect the mechanisms driving the spontaneous migration of living cells over a broad range of time scales, we followed single isolated MDCK epithelial cells (treated with mitomycin C to prevent cell division) on micro-contact-printed 1D linear tracks of fibronectin (Fig. 1a, b). This set-up allowed us to track cells from their early spreading phase and on over long time scales (48–96 h) using videomicroscopy (Fig. 1b, c). By detecting the cell edges, we could reconstruct the trajectories of single cells in absence of cell–cell contact interactions. We observed two main behaviours: a first population of cells exhibited static spreading, and extended slowly their two ends in opposite directions without net displacement of their centre-of-mass (Fig. 1d and Supplementary Movie 1), while a second population displayed strikingly regular oscillatory trajectories, with an amplitude that could significantly exceed the cell size (Fig. 1e and Supplementary Movie 2). While oscillatory patterns in cell migration[7,29,30], and more generally in cell dynamics[31–33], have been reported for various cell types, and could be attributed to different intracellular processes, we argue below that the oscillations that we observe have so far unrevealed features, which we show originate from a so far unreported mechanism. Both observed behaviours were approximately equally distributed over the cell population, whereas behaviours that did not fall in these two classes – akin to persistent random motion – remained negligible. These observations were qualitatively unchanged upon varying the width of the track over a range comparable to a single cell size, $W = 10, 20, 50\,\mu m$ (Fig. 1h and Supplementary Fig. 1a–c) and were recovered with cells that were not treated with mitomycin C, although in the latter case cells had to be tracked for shorter times (Supplementary Fig. 13).

To analyse the dynamics of front back cell polarity in these motility patterns, we used as a proxy for cell polarisation the concentration profile of a fluorescent biosensor (p21-activated kinase binding domain, PBD) of active Rac1 and Cdc42[24], which are two well-known activators of actin protrusions; as expected, moving cells displayed an increased PBD signal at the front, and lowered at the back, indicating their polarity. Two distinct phenotypes of polarisation, corresponding to the two observed dynamic behaviours clearly emerged from observations. On the one hand, static spreading cells were usually extended (up to > 100 μm) and characterized by a symmetric PBD profile with active poles at each of the two cell ends (see Supplementary Fig. 1g and Supplementary Movie 3). On the other hand, oscillating cells displayed 'run' phases characterized by reduced cell length (~20 μm), clear front back polarity, and roughly constant high speed (often faster than 100 μm. h$^{-1}$, see Fig. 1f, g, Supplementary Fig. 1 and Supplementary Movie 3, 4). These run phases were interrupted by phases of polarity reversal, where the cell phenotype was transiently comparable to that of static spreading cells.

We next focused on the oscillatory motility patterns, and characterized quantitatively their striking regularity. Despite some heterogeneity within the population, pointing to single-cell-specific properties, kymographs consistently displayed sawtooth-like patterns (Supplementary Fig. 1). Both the period $T$ and amplitude $A$ increased in time, starting from very short ($A \simeq 20\,\mu m$, $T < 1\,h$) scales and reaching very high values up to $A = 500\,\mu m$, $T = 20\,h$ (Supplementary Fig. 2). Strikingly, within single trajectories the $A/T$ ratio remained close to constant over this very broad range of values of $A$ and $T$. This was consistent with cells running at roughly constant speed and bouncing between two virtual reflecting walls imposing polarity reversals, which would slowly move apart. Based on this observation, and because in our set-up external cues imposing such dynamics could be excluded, we hypothesized that the observed polarity reversals were induced by interactions of cells with their own footprints, and not caused only by an autonomous intracellular clock. More precisely, we conjectured that cells modify the physicochemical properties of their local environment, thereby leaving long-lived footprints along their path. In turn, footprints were assumed to induce local polarity signals that favour cell polarisation pointing towards previously visited areas, and away from unvisited areas, and thus to effectively attract cells, thereby slowing down the large scale spreading of trajectories. In this scenario, cells would, therefore, run persistently while within the previously visited domain, and reverse their polarity when reaching an edge, thereby incrementally extending the visited domain by overshooting the edge.

**Cells deposit a footprint on their path.** To challenge this hypothesis, we prepared 'conditioned substrates' on which a first batch of cells plated at high density was left migrating freely; we thereby expected the substrate to be fully covered by cellular footprints. After removing this first batch of cells, we plated a new batch of isolated cells on these conditioned substrates (Fig. 2a), and performed the same analysis as in the control set-up (i.e. on substrates that were not conditioned by a first batch of cells). As compared to the control case characterized by slowly spreading oscillatory trajectories, cells on conditioned substrate displayed strikingly different migration patterns, with a drastically increased net displacement and a significantly larger persistence time, while only few oscillatory patterns could be observed (Fig. 2b, c, e, Supplementary Fig. 3 and Supplementary Movie 5). In addition, cell spreading at early times and cell instantaneous speed were found to be larger on conditioned substrates

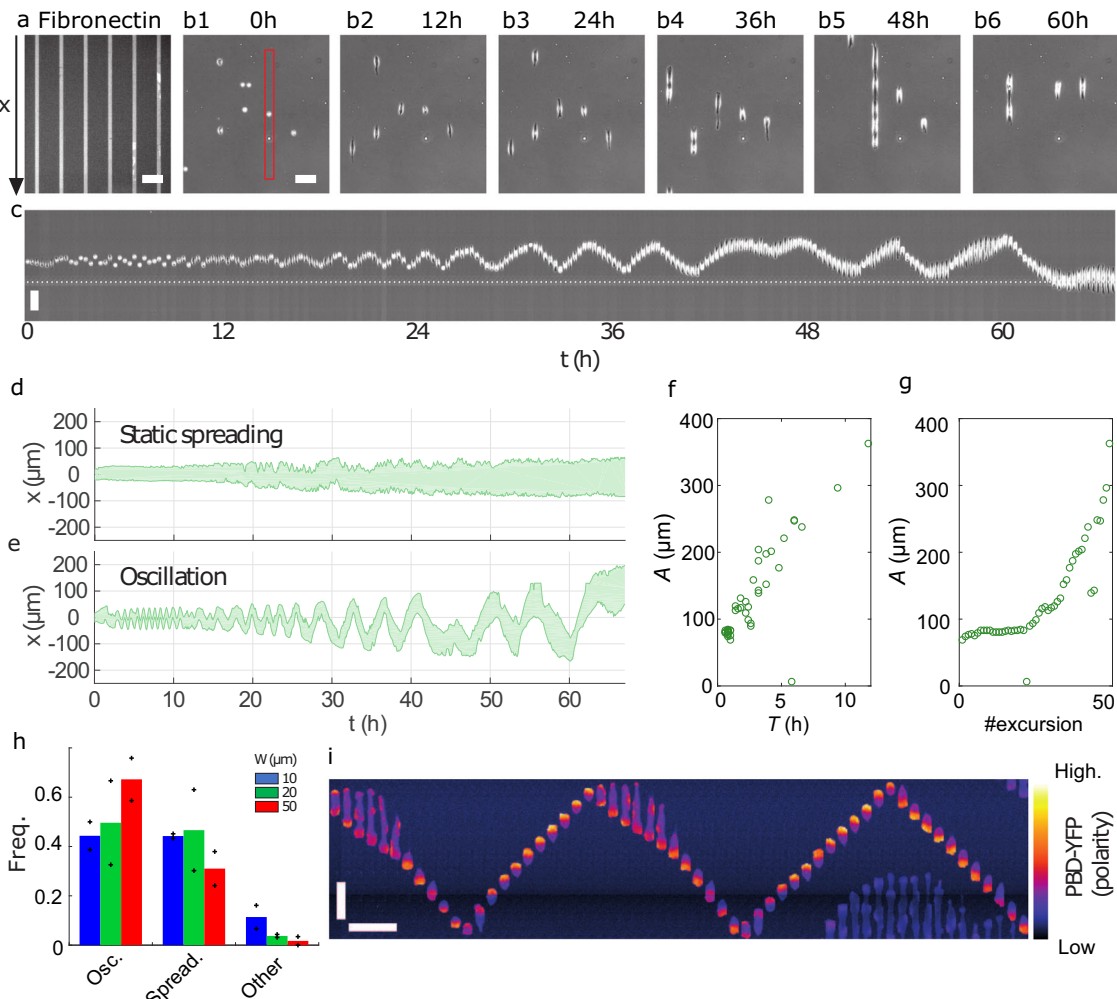

**Fig. 1 Isolated cells exhibit regular oscillations. a** Fluorescent tracks of fibronectin of width $W = 20\,\mu m$. **b1–b6** Snapshots of MDCK cells observed using phase-contrast imaging. **c** Kymograph of a single MDCK cell (red frame in **b1**) showing oscillations. **d, e**. Typical kymographs of a statically spreading (**d**) and an oscillating (**e**) cell plated on lines with $W = 20\,\mu m$. **f, g** Amplitude of the oscillations measured in panel **e** as a function of their period (**f**) and as a time series (**g**). **h** Frequency of the various behaviours (oscillating, spreading and other) of isolated MDCK cells on lines of different widths. $n = 61, 131$ and 87 trajectories for $W = 10, 20$ and $50\,\mu m$ from six independent experiments (2 per track width). Mean (bars) and single repeats (plus signs). **i** Kymograph of a MDCK cell expressing PBD-YFP, a reporter of Rac1/Cdc42 activation, hence of the cytoskeleton polarity. Individual frames are separated by 10 min. Typical behaviour observed in 2 independent experiments with PBD-YFP cells. All scale bars, 100 μm.

(Supplementary Fig. 3), indicating that cellular footprints facilitate adhesion and migration. This was further confirmed[34] by measuring the forces exerted by the cells on the substrate using traction force microscopy (TFM, Fig. 2f)[29]. In this 1D setting, the strength of the coupling between the cell and the substrate may be simply assessed by computing the maximal cell tension, which is obtained by integrating the 1D traction force profile along the line pattern (Fig. 2g, h). We observed that on average, cells on conditioned substrates were able to exert tensions twice as large as cells on control substrates (Fig. 2i). This might denote that on the edge of the footprint, the cell-substrate adhesions are weakened, hence decreasing the tension that they can withstand, possibly participating in the repolarisation of the cell motion. Importantly, these results are not limited to the 1D linear geometry. We repeated the motility assay on homogeneously coated surfaces and characterised the dynamics of free 2-dimensionnal cell trajectories: comparing substrates with and without conditioning by a first batch of cells, we observed a 4-fold increase of the effective diffusion constant on conditioned substrates as compared to control substrates (Fig. 2d, e). Finally, we challenged the generality of our observations by analysing two other cell types. First,

we found that Caco2—colorectal cancer—cells qualitatively reproduce the behaviour of MDCK cells: isolated Caco2 cells exhibit oscillations on control linear patterns, while on substrates conditioned by a first batch of Caco2 cells they move persistently (Fig. 2e and Supplementary Fig. 11). Second, we analysed the behaviour of RPE1 cells (Supplementary Fig. 12). On control substrates, oscillatory motion was again observed, but only for a subset of cells, while others moved persistently. Yet, in the case of oscillatory cells, we observed no noticeable increase of the amplitude of the movement with time, which leaves the door open for another origin of the oscillations in those cells. On conditioned substrates, our observations confirmed that cells moved—slightly but significantly—further than cells on control substrates. This confirmed the impact of cellular footprints on cell migration, even if in the case RPE1 the effect seemed milder and did not systematically lead to oscillations. Altogether, these results support our hypothesis of a generic phenomenon of cellular footprint deposition, which deeply impacts cell trajectories by restricting them to already visited areas.

Next, to substantiate our hypothesis, we sought to identify chemical components of the cellular footprints. It is known that

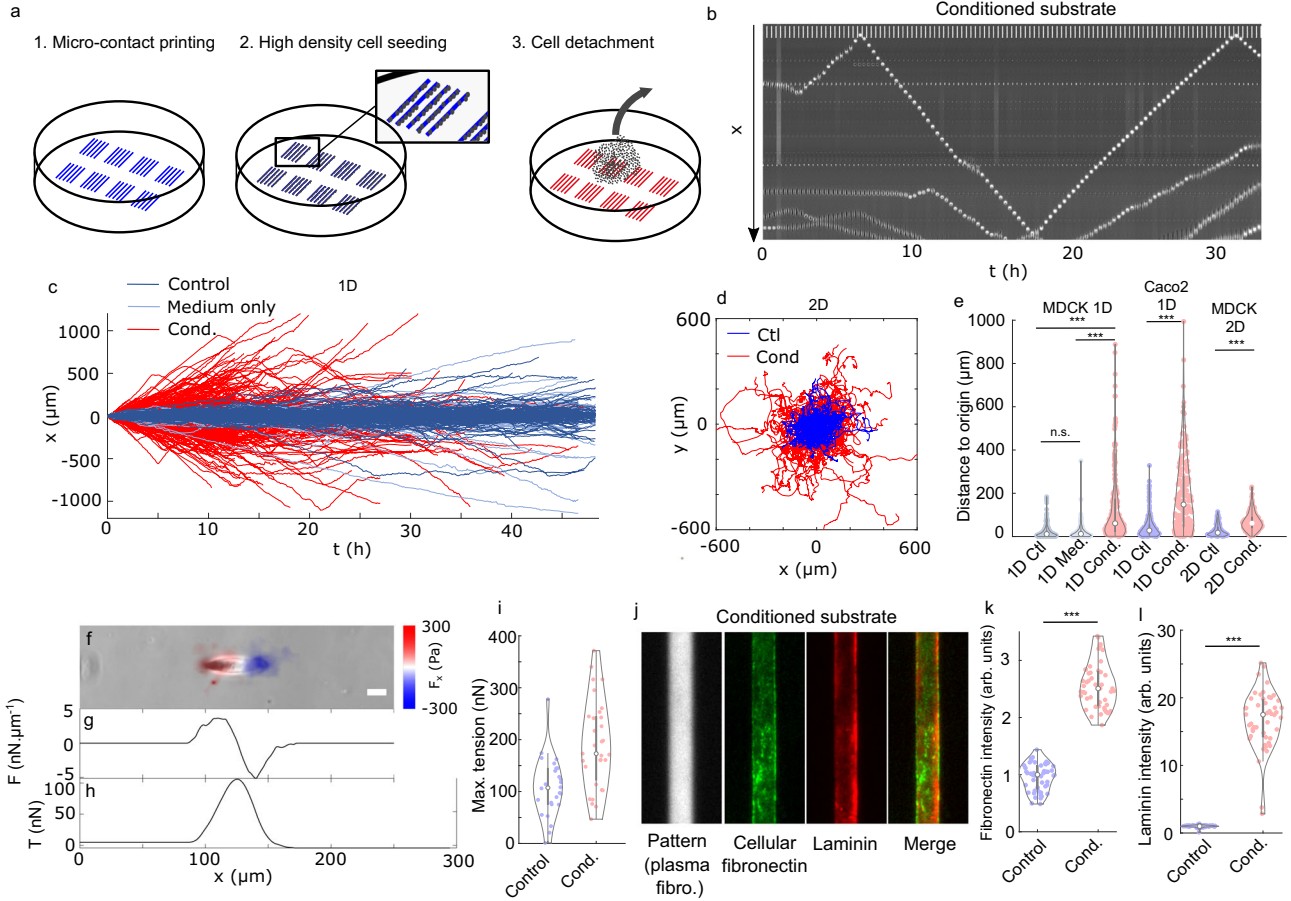

**Fig. 2 Cells deposit a footprint on their tracks. a** Principle of the substrate conditioning. Linear tracks were micro-contact printed (1), then a first layer of cells was plated at high density to recover all the surface (2) before being detached (3). **b** Kymograph of a cell moving along a conditioned 20 μm track with high persistence. Scale bar (repeated vertical line) 100 μm. **c** Trajectories of cells on control (Ctl: substrate kept in PBS, Medium only: substrate kept in DMEM during the same time as the conditioning) or conditioned susbtrates as a functions of time. This shows high persistence on conditioned susbtrates. Only trajectories of at least 10 h duration are shown. **d** Trajectories of single cells plated on 2D surfaces on control (Ctl, blue) or conditioned (Cond., red) substrates. **e** Distance of the cell to its original position after 16 h on control and conditioned substrates in 1D and 2D. Differences were assessed using the 2-sample Kolmogorov–Smirnov test, n.s.: non-significant ($p > 0.1$), ***$p < 0.001$. Exact $p$-values are $p = 0.89, 1.4e{-}29, 4.5e{-}26, 4.8e{-}27, 7.2e{-}24$ comparing between MDCK 1D (Ctl-Med, Ctl-Cond., Med.-Cond.), Caco2 1D and MDCK 2D data, respectively. $n = 355, 238, 429, 246, 216, 192$, and 194 trajectories from 3 (MDCK 1D, Ctl/Medium/Cond.), 3 (Caco2 1D, Ctl/Cond.) and 2 (MDCK 2D, Ctl/Cond.) independent experiments respectively. **f** Phase-contrast image of a MDCK cell on a conditioned line on soft PDMS, overlaid with traction stress. Scale bar 20 μm. Typical force profile observed in 3 independent TFM experiments with single MDCK cells. **g** Traction force profile along the cell, integrated over the line width. **h** Tension $T$ within the cell obtained by integrating the traction force profile along the $x$-axis. **i** Maximal (peak) tension of cells on control (blue) or conditioned (red) linear substrates. Difference between $n = 26$ and 31 cells was tested using the 2 sample Kolmogorov–Smirnov test, ***$p = 1.6 \times 10^{-4}$. **j** Fluorescence pictures of a line conditioned by MDCK cells, then fixed. 'Pattern' denotes the stamped labelled fibronectin, cellular fibronectin and laminin are immuno-stainings. **k–l** Cellular fibronectin (**k**) and laminin (**l**) staining intensity in control (blue) and conditioned (red) lines devoid of cells. Difference between $n = 48$ pictures (192 line patterns) from two independent experiments per condition was tested using the 2 sample Kolmogorov–Smirnov test, ***: $p = 5e{-}22$ for both (**k**, **l**). The violin plots in panels **e**, **i**, **k**, **l** display the full distribution of data points together with a standard box plot.

cells are able to assemble or produce extra-cellular matrix proteins, and in particular fibronectin[35], which thus appears as a natural candidate. Thus, we fixed both control samples after the patterning procedure, and conditioned samples after decellularisation. By immuno-staining, we showed that, on the conditioned substrates, the cells had deposited a subsequent amount of both fibronectin and laminin, which were absent from the control lines (Fig. 2j–l and Supplementary Fig. 16). In order to distinguish cell-produced fibronectin from pre-coated fibronectin, we used an antibody directed specifically to the fibronectin produced by the cells themselves[36]. Yet, the depletion of cell-produced fibronectin alone, using siRNAs, was not sufficient to suppress the oscillations (Supplementary Fig. 15), indicating that cellular

footprints were not defined by cell-produced fibronectin only. We also added marked plasma fibronectin in the medium, thus making it available for cells to capture, assemble and deposit along their path[37]. This strategy yielded similar results as the staining of cellular fibronectin, showing that cells are also able to deposit fibronectin that is available in the medium (Supplementary Fig. 10). Surprisingly, changing the initial surface density of fibronectin on the patterns did not affect the motion of cells, either on control or on conditioned substrates. Altogether, these results suggest that the deposition of ECM components by cells, and potentially their specific remodelling are key factors defining the cellular footprints, rather than the concentration only of a specific protein. We cannot exclude that other molecular or supra

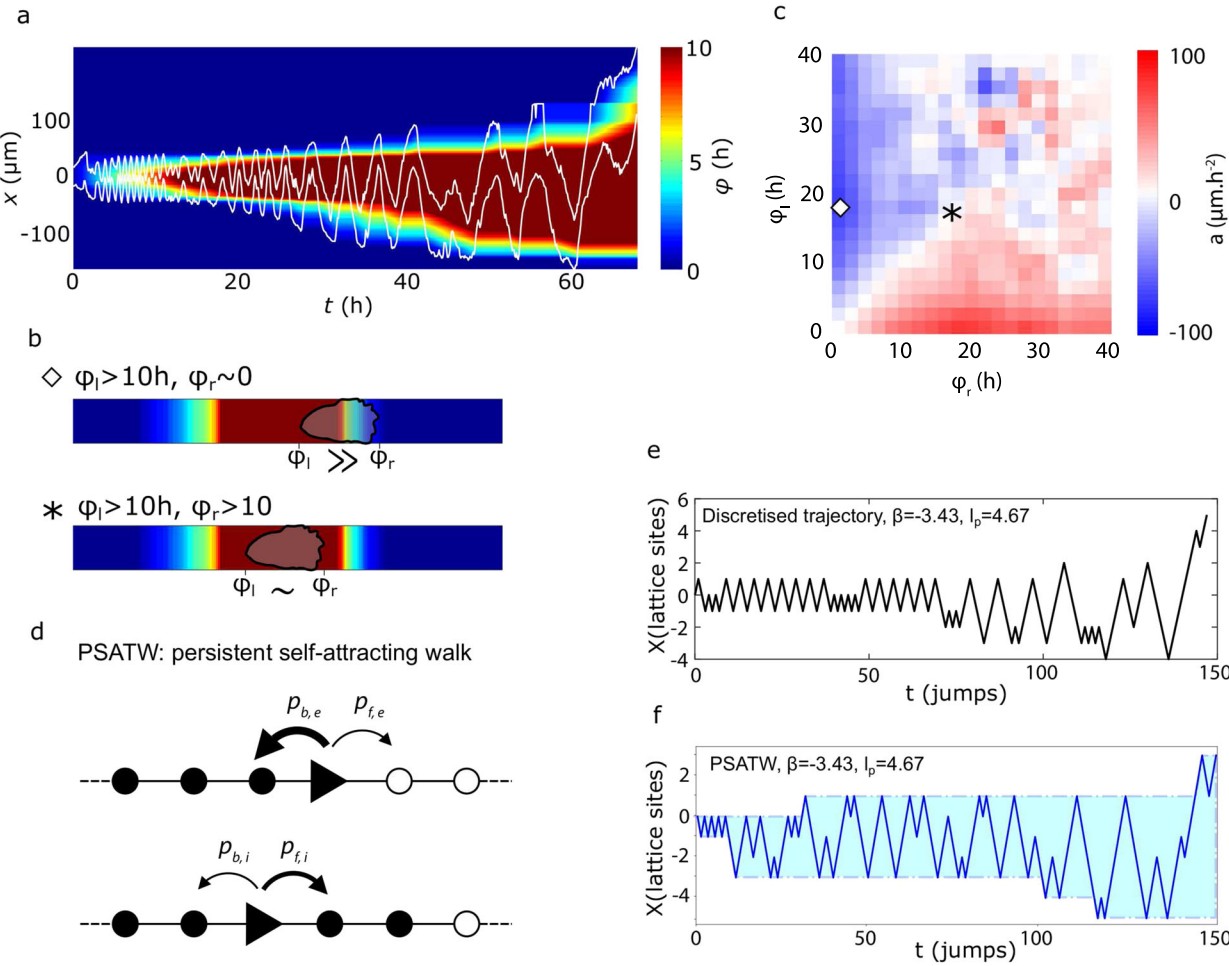

**Fig. 3 The trajectories of isolated cells have the characteristics of self-attracting walks. a** Kymograph of an isolated cell on a $W = 20\ \mu m$ track, overlaid with its footprint field $\varphi(x, t)$ defined as the cumulative time spent on a given position. **b** Sketch of the $\varphi_l$ and $\varphi_r$ measurements. Top: the cell sits on the right edge of the footprint, its right edge being outside with $\varphi_r \simeq 0$ while its left edge is within, $\varphi_l > 10\ h$. Bottom: the cell is completely within the footprint, with comparable high values of $\varphi_l$ and $\varphi_r$. The symbols are reported in panel **c** to show where each situations sits in the $\varphi_l - \varphi_r$ space. **c** Average acceleration of isolated cells on 20 μm tracks as a function of the values of $\varphi$ at both cell ends. This heatmap was made from 75,088 data points from $n = 131$ cell trajectories from two independent experiments. **d** Sketch of the persistent self-attracting walk model. Whether it is on the edge of its footprint (top) or in its interior, the walker have different probabilities to jump in the same direction as before or to turn back, set by two parameters $k$ and $\beta$. **e** Discretised experimental trajectory of an oscillating cell (same cell as panel **a**) allowing to measure the reversal statistics within and on the edge of the span. **f** Simulated trajectories of an agent following the PSATW dynamics with the same parameters as infered in (**e**).

molecular components, such as exosomes[38] or cell fragments[39] could be released as well, and even the sole remodelling of the pre-existing extracellular matrix could be invoked; a complete description of cellular footprints would go far beyond this work. However, our results provide a direct evidence that cells indeed leave long lived chemical footprints—made at least of fibronectin and laminin—which, as we showed, can deeply modify cell motion at later times.

**Cell trajectories as self-attracting walks.** To characterize the impact of cellular footprints on cell dynamics at the cell scale, we developed a kinematic approach based on the analysis of 1D cell trajectories. As a proxy for any potential deposited signal, we defined a footprint field $\varphi(x, t)$ as the cumulative time spent by a cell on a given location $x$ before time $t$ (Fig. 3a). We next analysed the correlations between the acceleration $a$ of the cell centre-of-mass, and the $\varphi$ values measured at the left ($\varphi_l$) and right ($\varphi_r$) ends of the cell (Fig. 3b, c and Supplementary Fig. 4–7). For a cell moving within the previously visited domain, the footprint field probed by the cell is roughly uniform ($\varphi_l \sim \varphi_r \gg 0$), and we

observed no significant variation of the cell migration speed ($a \sim 0$). In contrast, for a cell reaching for example the right (resp. left) end of the visited domain the local footprint field probed by the cell is very asymmetric with $\varphi_l \gg \varphi_r$ (resp. $\varphi_l \ll \varphi_r$), and we observed a significant average acceleration inward the visited domain, $\langle a \rangle \simeq -100\ \mu m\ h^{-2}$ (resp. $\langle a \rangle \simeq +100\ \mu m\ h^{-2}$). This clear correlation indicates that cell polarity is governed by local gradients $\delta\varphi = \varphi_r - \varphi_l$ of the footprint field, and substantiates our earlier hypothesis that cellular footprints impact on cell trajectories.

Our experimental results show that cells, by leaving chemical footprints along their way, are endowed with a spatial memory of their path. Their theoretical analysis, therefore, calls for a framework that goes beyond the classical models invoked in the literature, which are for most of them amenable to markovian, and, therefore, memoryless descriptions[5–7,15,17,40], with the exception of[41,42]. Our observations led us to argue that cell trajectories naturally fall in the class of self interacting random walks, which can be broadly defined as the class of random walks that interact (attractively or repulsively) with the full territory explored until

time $t$[43–50]. This class comprises in particular the self-avoiding random walk, which has played a crucial role in physics[51], and has applications in the modelling of trajectories of living organisms[52–54]. By construction, self-attracting random walks are endowed with long range memory effects, which makes their analytical study notoriously difficult.

A generic example of self interacting random walk is given by the so–called Self-Attracting Walk (SATW). It can be defined on a 1D lattice as a discrete time random walk whose jump probability to a neighbouring site $i$ is assumed to be proportional to $\exp(-\beta f(n_i))$, where $n_i$, defined as the number of times site $i$ has been visited by the random walker up to $t$, is akin to the footprint field $\varphi$ defined above. Upon varying the parameter $\beta < 0$ and the increasing function $f$ (case of self attraction), this model is known to display a broad range of behaviours, from everlasting trapping on a few sites to large scale diffusion[47]. More specifically we used the SATW to build explicitly a minimal model of cell dynamics that recapitulates our main observations on migrating cells. For the sake of simplicity, we took $f(n_i = 0) = 0$ and $f(n_i > 0) = 1$, which amounts to assuming that the deposited signal that defines cellular footprints is bounded. Next, we extended the SATW model to take into account cell persistence. The persistent self-attracting walk (PSATW) can be defined as follows in 1D. When the walker is on a site $i$ within the visited domain— i.e. surrounded by sites that have already been visited, $n_{i-1}, n_{i+1} > 0$— it performs a classical persistent random walk: it changes direction with probability $p_{r,i} = \frac{e^{-k}}{e^{-k}+e^{k}}$, and reproduces its previous step with probability $1 - p_{r,i}$, where $k > 0$ is a parameter that controls the cell persistence length $l_p = e^{2k}$ (the persistence time $t_p$ is defined identically by setting cell speed to 1). When the walker is at an edge of the domain (eg $n_{i+1} > 0$), it experiences a local bias inward the visited domain parametrized by $\beta < 0$ and the probability to change direction can be written $p_{r,e} = \frac{e^{-k-\beta}}{e^{-k-\beta}+e^{k}}$, while the probability to reproduce the previous step is $1 - p_{r,e}$ (Fig. 3d). With this definition, a typical PSATW trajectory with $k > 0$ and $\beta < 0$ shows noisy oscillatory patterns, with an amplitude that slowly increases over time, which qualitatively reproduce experimental observations (Fig. 3e, f). More quantitatively, we found that the experimental trajectories, after adequate discretization, could be well fitted by adjusting the $k, \beta$ parameters, with an inherent cell to cell variability (Supplementary Fig. 8). Of note, the fitted $k$ values, which control the intrinsic cell persistence length, were comparable on conditioned and control substrates, while the parameter $\beta$, which controls the cell response to the footprint field, was significantly different in both conditions (Supplementary Fig. 9). This finally shows that PSATWs provide a minimal model of the self interacting random walk class, which reproduces the observed migration patterns.

**Long-term consequences of spatial memory.** Last, we show both theoretically and experimentally that the reported interaction of cells with their footprint, which endows cells with a memory of their path, has important consequences on space exploration properties of cell trajectories. (i) First, the time dependent increments defined by $I(T,t) \equiv \langle [x(t+T) - x(T)]^2 \rangle$, which quantify the spreading speed of trajectories, are found to depend on the measurement time $T$ at all time scales, ie to display ageing, in both 1D and 2D set-ups and in agreement with the 1D PSATW model and the 2D SATW model (Fig. 4a, b, d, e—note that cell to cell variability and fluctuations of cell speed were taken into account in the numerical simulations of the 1D PSATW model, see SI). Conversely, we observed that ageing of the increments was negligible on conditioned substrate (Fig. 4c,f), further confirming our findings that memory effects where induced by cellular footprints. This is a direct consequence of the increase over

time of the span of the visited territory. In 1D, at short measurement times $T \ll t, t_p$, the increments dynamics is governed by interactions events with the edges of the visited domain, which slow down spreading and lead to a diffusive behaviour $I(T,t) \sim t$ for both $t < t_p$ and $t > t_p$ (for $\beta$ large enough). For $T \gg t, t_p$, edge effects become negligible and one recovers the classical dynamics of persistent random walks, which crosses over from a ballistic ($I(T,t) \sim t^2$) to a diffusive ($I(T,t) \sim t$) regime. In 2D, the observed persistence length is comparable to the cell scale and can be neglected; one can thus use a classical SATW model. This model was shown to lead to normal diffusion in the $T \gg t$ regime, and to subdiffusion[47,48] $I(T,t) \sim t^{2/3}$ in the $T \ll t$ regime, which is consistent with our observations, even if experimental data do not allow to determine quantitatively the exponent. This subdiffusive regime shows that memory effects can have drastic consequences on space exploration, by changing the very dimension of trajectories, which qualitatively become more compact. (ii) Second, we argue theoretically that such ageing dynamics has important consequences on first-passage time statistics[55,56], which is a key observable to quantify the efficiency of migratory patterns to find 'target' sites of interest in space[57,58]. In the simplest theoretical setting of a single target located in infinite space, first-passage statistics to the target are conveniently parametrized by the persistence exponent $\theta$, which defines the long time asymptotics of the survival probability of the target $S(t) \propto t^{-\theta}$[59]. For a broad class of random walks, which do not display ageing, the persistence exponent takes the remarkable universal value $\theta = 1/2$[59]. Strikingly, our results show that memory effects in the PSATW model in 1D lead to non trivial values of $\theta$[56], which are controlled by the parameter $\beta$ ($\theta = \frac{e^{-\beta}}{2}$, Fig. 4c). Of note, one has $\theta > 1/2$, indicating that memory effects increase the relative weight of shorter trajectories and thus favour local space exploration, as compared to memoryless random walks with $\theta = 1/2$. First-passage statistics are thus deeply impacted by memory effects in the PSATW model.

**Discussion**

Moving cells interpret multiple physical ECM parameters in parallel and translate them into an integrated response, which determines cell shape, polarity and migration[60]. From this well-established principle, our results provide further steps in our understanding of cell-matrix interactions. We indeed demonstrate that the reciprocal dynamic adaptation between the cell and its external environment is crucial to determine cell migration principles. Cells not only respond and adapt to their environment, but they are also able to build their own road while advancing. Here we show how cells keep track of their previous locations by depositing a footprint on their path, and, in turn, how this footprint determines dynamical properties of cell migration. Our discovery of self-attraction mechanisms during cell migration manifests through the emergence of oscillatory modes when cell motion is confined on a 1D line and peculiar sub-diffusive trajectories over 2D surfaces.

In fact, we anticipate that these feedback mechanisms between cell migration and substrate modifications could play an important role in many other situations where it was not suspected. In vivo, motile cells integrate various inputs from the extra-cellular matrix, and, then, adjust their migration mode[61]. Gagné et al.[62] showed that ILK potentiated intestinal epithelial cell spreading and migration by allowing fibronectin fibrillogenesis. The assembly of new roads built up by the cells themselves can be also crucial in the context of collective cell migration where the remodelling of ECM by leader cells can provide a path for follower cells[63]. Such retroactions between intra- and extra-cellular dynamics could thus be a more generic feature than it is currently

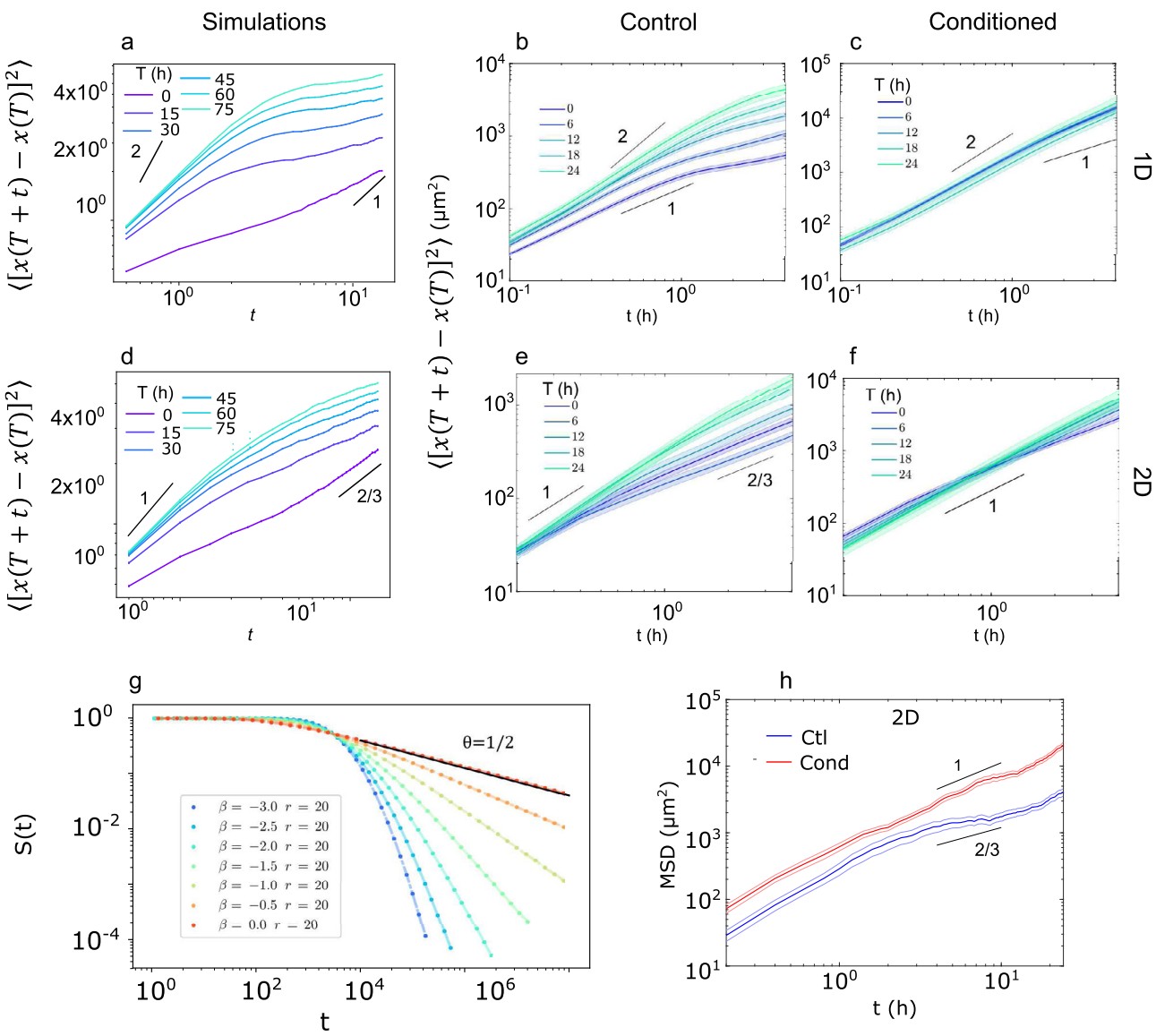

**Fig. 4 The PSATW model predicts altered cell trajectory statistics and a better exploration of space in both 1D and 2D. a–f** Increments of the mean square displacements in 1D (**a–c**) or 2D (**d–f**) settings. Data from simulations (**a, d**) and experiments on control (**b, e**) and conditioned (**c, f**) substrates. Lines are guides for the eye showing the various exponents predicted by the theory. In panel **a** cell to cell variability and fluctuations of cell speed were taken into account as discussed in SI. In panels **b, c, e**, and **f** the error bars show the S.E.M. for $n > 100$ trajectories from 3 (1D) and 2 (2D) independent experiments. **g** Survival probability $S(t)$ of a target located at a distance $r$ of the origin in 1D as a function of time. The long-time scaling exponent, $\theta$, decreases as $\frac{e^{-\beta}}{2}$. **h** Mean square displacement of isolated cells moving on 2D surfaces in logarithmic scale. Control (Ctl, blue) and conditioned (Cond., red) substrates. The error bars show the S.E.M. for $n = 203$ and 172 trajectories from two independent experiments for control and conditioned substrates, respectively.

considered and previous works on cell migration might be revisited in the light of our findings. In the present case, the self-attraction mechanism described here has three main consequences that might be of biological significance. First, it generates loose self-confinement, which at first sight prevents efficient migration over long distances. In return, this localisation ensures a much better exploration of space, so that no unexplored 'hole' is left behind, which might be of importance for cells that need to patrol a zone. Finally, it confers ageing on cell trajectories, which might be crucial although overlooked in cell migration experiments: it means that the movement properties at a given time can depend strongly on the interval between the time at which cells are deposited and the time at which measurements start. Two questions remain open: the nature of the footprint and the physical mechanism by which the cell is attracted back.

Several works have shown that cells are able to produce and to remodel their extra-cellular matrix[35,37,62,64–66]. In this study, we have shown that cells actually deposit fibronectin in the form of small puncta but other ECM components may play a role. In addition, other candidates have been evidenced as self-attraction media, notably exosomes[38]. The mechanism of repolarisation at the footprint edge also remains unclear, although its details could be of importance for the overall dynamics: the cell-substrate system needs to be close to a specific operation point, such that the footprint is deposited efficiently enough for the cell to sense it, but not too fast so the cell reverses its polarity before it has built a strong enough footprint at its front to keep moving ahead.

Our findings provide a framework to understand the intimate relationship between cell and ECM remodelling. Even though our study focuses on single cell behaviour, we anticipate that it could

play a role in collective cell dynamics. In living tissues involving cell populations, either sparse or dense, either homogeneous or heterogeneous, reinforced motion could manifest through a broad variety of consequences. For instance, Attieh et al.[66] showed that cancer associated fibroblasts could open the way to cancer cells by assembling fibronectin fibrils along collagen fibres. There is no doubt that such amazing collective effects arise when several self-attracting walks interact, or if the self-attracting field can be degraded with time. Future studies may try to introduce those levels of complexity and analyse the role they play in various physiological and pathological situations.

## Methods

**Cell culture**. We used MDCK wild-type, MDCK histon-GFP, MDCK PBD-YFP (gift from F. Martin-Belmonte lab), Caco2 and RPE1 cells. The cells were cultured in DMEM GlutaMAX high-glucose (Gibco, Waltham, MA) supplemented with 10% foetal bovine serum (FBS, BioWest, Nuaillé, France) for MDCK lines, and 20% FBS + 1% penicillin–streptomycin for Caco2. RPE1 cells were cultured in DMEM/ F12 (Gibco) supplemented with 10% FBS. Prior to experiments, the cells were treated with 10 µg. mL of mitomycin C added in the medium for 1 h, then rinsed before subsequent detachment and seeding on the experimental samples.

**siRNA depletion of fibronectin**. Caco2 cells were transfected with siRNA targeting the human fibronectin RNA or with non-targeting siRNA (Dharmacon smartpools) using lipofectamin 2000 (Invitrogen) following the manufacturer's instructions. Cells were transfected twice at 24 h interval, and they were seeded on lines at $t = 72$ h after the first transfection. In parallel, cells were lysed at the same time (72 h) for western blot analysis of their fibronectin content.

**Micro-contact printing**. All micropatterns were prepared using standard micro-contact printing on PDMS, as described previously[29]. The substrates used were non-culture treated plastic dishes (Greiner Bio-One, Kremsmünster, Austria) for experiments in Fig. 1, glass coverslips (Menzel-Gläser) for PBD-YFP experiments and 6-well plates for all other experiments. The substrates were first covered with a thin layer of poly-dimethyl-syloxane (PDMS, Sylgard, Dow Corning, Midland, MI)) using a spin-coater and crosslinked at 80 °C for 2 h. PDMS stamps were prepared by pouring PDMS on a mold featuring the patterns to be printed and crosslinked as described. After cooling down, a fibronectin solution was prepared by adding 5 mg mL⁻¹ of fibronectin and 2.5 mg mL⁻¹ of Cy3- or Cy5-labelled fibronectin into sterile milliQ water. The solution was then incubated on the stamps for 40 min at room temperature. Before stamping, the substrates were activated using UV-ozone for 10 min and the stamps were rinsed to remove excess fibronectin and dried using an air-gun. The stamps were briefly put in contact with the substrate's surface, then removed, and the substrates immerged in a 2% pluronics F127 (Sigma-Aldrich, Saint Louis, MO) solution in PBS for 2 h. Finally, the substrates were rinsed in PBS and sterilised under the UV lamp of a culture hood before use. The substrates for 2D migration were prepared similarly, using a large piece of flat PDMS as a stamp for homogeneous coating. For the data in Supplementary Fig. 14, various fibronectin concentrations were used in the first—adsorption—step.

**Substrate preparation for traction force microscopy**. For traction-force microscopy (TFM) experiments, soft PDMS substrates were prepared and micro-contact printed as described previously[29]. Soft PDMS with Young's modulus of approximately 30 kPa was prepared by mixing components A:B of soft PDMS in a 5:6 weight:weight ratio, then poured on a 6-well plate and cured at 80 °C for 2 h. Fluorescent beads (Fluorospheres, Invitrogen, Carlsbad, CA) were stuck on the surface after coating with aminopropyl-triethoxy-silane (APTES, Sigma, Saint Louis, MO). Finally, the substrate was patterned by first using micro-contact printing on a poly-vinyl-alcohol (PVA, Sigma, Saint Louis, MO) membrane. Then the membrane was put in contact with the substrate for 30 min, dissolved in water and rinsed before passivation with Pluronic F127 as above.

**Conditioned substrates**. To prepare conditioned substrates, substrates were first prepared as above. Conditioning cells were seeded at high density so as to cover all the available surface, rinsed after 45 min to prevent overcrowding and placed in the incubator. After 12 h of conditioning, the good covering of the substrate was checked under the microscope, and the culture medium was replaced with a 20 mM solution of EDTA in calcium- and magnesium-free PBS (Gibco, Waltham, MA). After 2 h, the remaining cells were detached by flowing gently the solution using a pipette, and the substrate was rinsed with PBS.

**Cell deposition**. Cells were detached using Trypsin-EDTA (Gibco, Waltham, MA), counted, centrifuged, resuspended in a 1:1 mix of full culture medium and low calcium DMEM to prevent too much cell clustering. Typically $10-20 \times 10^3$ cells were seeded per well in control experiments on lines, while only $1 \times 10^3$ was

enough for experiments on conditioned substrates (even lower densities were used for experiments on 2D surfaces). Precise control of the cell density was necessary to ensure good statistics with a high number of cells remaining isolated for a long duration. After 45 min, the sample was rinsed in PBS with great care to remove most floating cells without affecting recently adhered cells and 3 mL of full culture medium was added.

**Live imaging**. Live experiments were performed using a fully automated inverted microscope (Olympus, Japan) with computer-controlled temperature and $CO_2$ level. At least one fluorescent image of the patterns was taken at the beginning of the experiment. Cells were imaged using phase-contrast imaging, with a time-lapse of 6 (1D), 10 (TFM) or 12 (2D) minutes between frames. For 2D experiments, histon-GFP expressing cells were used and imaged in fluorescence to make cell detection easier. For TFM experiments, the focus was done on the substrate's surface and fluorescent images of the beads were acquired at each frame. At the end of the experiment, the cells were detached using Sodium dodecylsulfate (SDS, Sigma, Saint Louis, MO) at 10% and an image of the beads was taken as a force-free reference. The imaging of PBD-YFP was done using an inverted microscope (Leica, Wetzlar, Germany) with a CSU-W1 confocal spinning-disk module (Nikon, Tokyo, Japan) and a 40X oil-immersion objective. The acquisition was done using Metamorph (Molecular Devices, San Jose, CA), at a 6–10 minutes acquisition rate. The focus was done on the basal plane of the cells and the microscope's hardware autofocus was used to ensure the absence of defocusing.

**Labelled fibronectin deposition**. To check for the deposition of fibronectin from the medium by the cells, 5 µg mL⁻¹ of Cy3- or Cy5-labelled fibronectin was added in the medium just before imaging – choosing the colour in order to avoid confusion with the micro-contact-printed fibronectin. The sample was imaged, either in PBS or mounted in Moewiol, after fixation and potential complementary immuno-staining.

**Immuno-staining**. The sample was fixed in 4% paraformaldehyde for 15 min, then rinsed with PBS. The primary antibody (anti-[IST9] fibronectin, AbCam ab6328) was added at 1/100[66] and left at 4 °C for 48 h, then rinsed with PBS. Laminin staining was performed using Sigma-Aldrich anti-laminin produced in rabbit (L9393) at 1/100. The secondary antibodies—Alexa 488, goat anti-mouse (Life Technologies A11001) and Alexa 568, goat anti-rabbit (Life Technologies A11011) —were added at 1/250 and left to incubate at room temperature for 2 h, then rinsed with PBS. Finally, the sample was either placed in PBS (for 6-well plates) or mounted on a glass slide with Moewiol.

**Image analysis—cell migration**. The image treatment was performed using homemade ImageJ (FIJI distribution) and Matlab (The Mathworks, Natick, MA) software. In brief, patterns were first detected using semi-automated methods, images were rotated so as to align all patterns on the same direction and positioning noise was removed if necessary using `Template Matching` ImageJ plugin. Cell contours were detected using the Edge detection function of ImageJ and thresholding with filters on the detected features' size and circularity, and the positions of both cell ends were recorded. For 2D experiments, the nucleus centre-of-mass was detected using an in-house program based on the `Find Maxima` function of ImageJ[17]. The cells were tracked in time using the `track.m` (http://site.physics.georgetown.edu/matlab/) program in Matlab to reconstruct full cell trajectories. The trajectory data were organised as in[17] and adapted software was developed for all further analysis (footprint, velocity, auto-correlations, oscillation detection, MSDs...).

**TFM analysis**. The TFM data were analysed as previously described[29]. First, the displacement field of the beads was measured using particle-image velocimetry (PIV), using the `matpiv` (https://www.mn.uio.no/math/english/people/aca/jks/matpiv/) function in Matlab. Then the force field was computed using the `FTTC`[67] program in ImageJ (https://sites.google.com/site/qingzongtseng/tfm). For each time frame, the cell mask was extended so as to engulf all the corresponding force patch, and the force profile along the cell length was computed by averaging the forces within this mask along the line direction. The 'total force' is the integral of the absolute values of this profile and hence corresponds roughly to twice the peak tension within the cell. For each cell, the median total force was calculated over the experimental time.

**Fluorescent fibronectin analysis**. To measure the intensity of fluorescence of immuno-stained cellular fibronectin, we proceeded as follows. First, pictures were taken with the wide-field microscope to get quantitative data. The issue of this imaging technique is that the illumination field is very inhomogeneous. To remove this bias, we sought to measure locally the difference of fluorescence between the inside and the outside of the line patterns. To that end, we defined local ROIs of 130 µm length along the line and 60 µm width centred on the line centre. We computed the intensity profile in the region outside the line, averaged over the ROIs length along the line. This profile was fitted with a linear function and the intensity profile corrected so that the intensity outside the line was fluctuating

around 0. Finally, the intensity at a given location along the line was computed as the average of this corrected intensity over the line width. The location of cells on the lines was detected by simply thresholding the histon-GFP intensity. Then for all lines completely devoid of cells, we computed the median of the corrected intensity profile. For lines with cells, the corrected intensity was binned as a function of the distance to the closest cell nucleus, and the median was taken for each distance bin to show the decay of fluorescence away from cells.

**Analysis of immuno-stained conditioned substrates**. The location of the line patterns was detected by thresholding on the pattern channel. Then, for each picture, the median of the fluorescence intensity in both fibronectin and laminin channels was measured both inside and outside of the line, and then the difference between the values inside and outside. Finally, the values were normalised by dividing by the mean value of fluorescence intensity in control substrates.

**Discretisation procedure**. To analyse cell trajectories in the framework of the PSAW model, we discretised the experimental trajectories as follows. The centre-of-mass position $x$ was interpolated on a regular grid $X = \lfloor \frac{x}{L_{ref}} + 0.5 \rfloor$ whose mesh size $L_{ref}$ was defined as a cell-specific typical length. Then the time was interpolated on the irregular grid $(T_i)_{i \in \mathbb{N}}$ corresponding to hopping events $X(T_i) = X(T_i - 1) \pm 1$. Note that this way, time is irregularly interpolated. The distribution of jump times $(t_j^i) = (T_{i+1} - T_i)$ is bi-exponential with a mean of $\langle t_j \rangle \simeq 0.6$ h, yielding discretised experimental trajectories of typically 100–150 time-steps (Fig. 3f of the main text). Directly measuring the conditional frequencies of reversal $p_{r,i}$, and $p_{r,e}$, we computed one $(k, \beta)$ parameter set per cell by inverting the expressions of those probabilities in the PSAW model formulation.

**MSD and ageing**. The MSD plotted in Fig. 4h is simply computed for each cell as $\|\mathbf{x}(t) - \mathbf{x}(0)\|$. For the ageing of the MSD, the trajectories were first cut in 12 h-long pieces starting at $T \in \{0, 6, 12, 18, 24\}$ h. The MSD was computed as $\mathrm{MSD}(T, t) = \langle \| \mathbf{x}(t + t_0) - \mathbf{x}(t_0) \| \rangle_{t_0}$, where the angle brackets denote a sliding average over $t_0 \in [T; T + 12$ h] to ensure better statistics, and the curve was cut at $12/3 = 4$ h for the same reason.

**Statistics and plotting**. To test the statistical significance of the difference between data distributions, we used the two-sample Kolmogorov–Smirnov test implemented in Matlab. Throughout the manuscript, to describe the data distributions accurately we used violin plots generated with the `violinplot.m` function in Matlab[68]. Briefly, this data representation includes all the data points as a scatter, superimposed with a smoothed (vertical) histogram of the data and a box-and-whisker plot displaying the median, the lower and upper quartiles and the minimum and maximum of the distribution. The figures were organized and prepared in Inkscape.

**Reporting summary**. Further information on research design is available in the Nature Research Reporting Summary linked to this article.

## Data availability
Data supporting the findings of this manuscript are available from the corresponding authors upon reasonable request. Source data are provided with this paper.

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

## Acknowledgements
We thank the group members from 'Cell adhesion and migration' team for helpful discussions. This work was supported by the LABEX Who Am I? (ANR-11-LABX-0071), the Ligue Contre le Cancer (Equipe labellisée 2019), and the Agence Nationale de la Recherche ('POLCAM' (ANR-17-CE13-0013 and 'MechanoAdipo' ANR-17-CE13-0012). We acknowledge the ImagoSeine core facility of the IJM, member of IBiSA and France-BioImaging (ANR-10-INBS-04) infrastructures.

## Author contributions
Jd.A., B.L. and R.V. designed the research. Jd.A. performed the experiments and analysed the experimental data, except TFM experiments, which were performed and analysed by V.C. A.B.C., O.B. and R.V. designed the model. A.B.C. performed the numerical simulations and analysed the simulation data. O.B., R.M.M., R.V. and B.L. supervised the project. Jd.A., A.B.C., B.L. and R.V. wrote the manuscript. All authors commented on the manuscript and agreed on its final version.

## Competing interests
The authors declare no competing interests.
