## [Peer Review File · Nature Communications]

Reviewers' Comments:

Reviewer #1:

Remarks to the Author:

The manuscript by d'Alessandro, Barbier-Chebbah et al. reports on experiments and theoretical work done to investigate how cellular migration is affected by deposition of extracellular matrix. Using 2D micropatterns, it is demonstrated that cells leave footprints, in particular fibronectin, along their way, which determines future migration of the same cell on this pattern. Mathematically, the stochastic motion of cells is described as self-attracting random walks that exhibit anomalous subdiffusive behavior.

This manuscript addresses a fundamental question, namely, how extracellular matrix deposition can generate spatial memory that affects the migration patterns of cells. The study is well-designed and has been executed carefully. The combination of thoughtful experiments and a transparent mathematical model is appealing, both to biologists and physicists. Moreover, the manuscript is carefully written, very clear, and will therefore also be of interest for non-specialists. Therefore, the reviewer recommends publication of the work after an issue with the mathematical model has been fixed (see below) and some optional minor improvements have been made.

The following points should be addressed:

-Figure 4a) The incremental msd curve from the PSATW simulations shows a maximum which is absent in the experimental data. This very pronounced difference between simulation and experiment should be explained and the model should be improved to fully capture the measurement results.

-Title: The title "Cell migration driven..." is a bit misleading since the migration is driven by internal mechanisms while the authors discuss here extracellular guidance cues. One could instead write "Cell migration guided..."

-The authors mention that a subpopulation of the cells only spread while others undergo oscillatory motion. Can the authors elaborate on this finding? Why is this happening? What is the role of inhomogeneities in the printed contacts?

Line 243: a pronoun is missing in "...reciprocal dynamic adaptation between cell and its..."

Reviewer #2:

Remarks to the Author:

See my attached file.

Henrik Flyvbjerg

Report on NCOMMS-21-01416-T: Cell migration driven by long-lived spatial memory

January 26, 2021

1 Summary

This manuscript presents results from a very thorough investigation of a subject that should have been addressed long ago by researchers studying cell motility on surfaces. The results presented combine rich experimental data with theoretical insights via a solid data analysis and computer simulations. This is science done very well. The high quality of this work and its presentation is obviously the result of a very close collaboration between experimenters and theorists—this is admirable. The manuscripts presentation is clear, in fairly perfect English.

Conclusion: Publish after authors have considered my first item below and corrected the typos listed in the ensuing items.

2 Details

1. Line 55: The references given here for oscillatory patterns in cell migration is nice. But they are from 2014 and 2019, and a much more striking observation of the phenomenon was published earlier, in 2011 in Ref. 13: Its Figures 4, 6, and 7 show distinct peaks/bumps in the power spectrum of velocities of individual, non-interacting amoebae, clear indicators of an oscillatory component (with drifting phase) to the self-propulsion of these single-cell organisms. Velocity auto-covariances shown in Ref. 13 show the same, thus confirming what cell trajectories

shown in Ref. 13 seem to show: These single-cell organisms march to a rhythm of their own, though the phase of this rhythm drifts rather fast.

2. Lines 198–199: the fitted k values, that control the intrinsic cell persistence length, were ... → the fitted k values, which control the intrinsic cell persistence length, were ... [if there is only one kind of k -values] OR the fitted k values that control the intrinsic cell persistence length were ... [if there are more than one kind of k -values.]
3. Line 279: tough → though
4. Figure 4, caption: “**b.**” → “**g.**” and “**d.**” → “**h.**”
“The long-time scaling θ exponent decreases ...” → “The long-time scaling exponent, θ , decreases ...”
5. Line 289: See LaTeX manual for how to make correct quotation marks.
6. Reference 14: Bruckner → Brückner
7. Reference 19: Bruückner → Brückner
8. Supplementary Fig. 2: Panels b and c of the figure have switched names in the caption.
9. Supplementary Fig. 3: This figure is not mentioned/referred to anywhere in Supplementary Material. But the article does refer to it. Its caption should explain the strange-looking error bars of the figure.

Reviewer #3:

Remarks to the Author:

Cells interact during locomotion with their microenvironment. During this interaction, cells do not only sense cues (chemical, mechanical, geometrical) in the extracellular matrix (ECM), but also remodel the ECM. Remodelling includes mechanical ECM rearrangement, ECM digestion (by e.g. secreting proteases), and ECM deposition (e.g. fibronectin). Deposition of ECM components by cells is well known to influence diverse biological aspects (e.g. Helvert, S. van, Storm, C. & Friedl, P. Mechanoreciprocity in cell migration. *Nature cell biology* 20, 8–20 (2018); Humphrey, J. D., Dufresne, E. R. & Schwartz, M. A. Mechanotransduction and extracellular matrix homeostasis. *Nature Reviews Molecular Cell Biology* (2014)).

d'Allessandro et al use patterned 1D lines - a well established method to analyse cell migration in a highly reductionist manner - and observe an unexpected oscillatory movement of MDCK (canine kidney) and Caco2 (colorectal cancer) cells. Monitoring this oscillatory movement for up to 96 hours revealed a continuously increasing length of the cellular path. By experimentation and theoretical modelling, the authors can convincingly show that this phenomenon can be explained by cellular deposition of ECM material (such as fibronectin) onto the oscillatory cellular paths and importantly onto the boundaries of the oscillatory cellular paths, thereby continuously increasing the length of the cellular path. Whereas the experiments are largely convincing, the significance/relevance and some of the interpretations (cellular memory; space exploration) are questionable/unclear.

Major points:

Significance: several publications investigated migration on 1D lines. However, oscillatory cell movement on 1D lines has been so far only reported upon cellular manipulation, including zyxin-depletion (focal adhesion-/ actin stress fibre-protein; Fraley, S. I., Feng, Y., Giri, A., Longmore, G. D. & Wirtz, D. Dimensional and temporal controls of three-dimensional cell migration by zyxin and binding partners. *Nat Commun* 3, 719 (2012)) and microtubule-inhibition (Zhang, J., Guo, W.-H. & Wang, Y.-L. Microtubules stabilize cell polarity by localizing rear signals. *Proceedings of the National Academy of Sciences* (2014)), or upon usage of very defined ECM/pattern boundaries (Zhou, F. et al. Quasi-periodic migration of single cells on short microlanes. *PLoS ONE* 15, e0230679 (2020).; Brückner, D. B. et al. Stochastic nonlinear dynamics of confined cell migration in two-state systems. *Nature Physics* 15, 595–601 (2019)). In contrast, d'Allessandro et al observe oscillatory cell movements without cellular manipulation or defined pattern boundaries. This raises the question, whether the authors are looking at a very specific situation, in which the adhesiveness of the pattern (or of the particular cell lines) are just as adhesive to allow cellular binding to the pattern, but not adhesive enough to enable persistent cell migration without additional cellular ECM deposition.

Interpretations: the title ("driven by long-lived spatial memory") and the abstract ('cells remember their paths') could misleadingly imply that cells actively (and potentially intracellularly) generate a memory. Yet, the authors show in this manuscript that the extracellular deposition of ECM material (fibronectin) and subsequent binding to the ECM material influences the kinetics of cell migration. Their preconditioning experiment demonstrate that other cells than the depositing-cell can respond (as one would expect) to the deposited ECM - thus the findings are rather about deposition/detection cycles than about a memory.

- Interpretations: the observed phenomenon rather confines cellular space exploration, as MDCK and Caco2 cells only increase their path length for short distances during every oscillation cycle. Thus, a title "Cell migration confined by.." instead of "Cell migration driven by.." could more accurately reflect the findings of the authors. Similarly, the authors should clarify this aspect better in the abstract and discussion.

Further aspects:

Line 123: "generic phenomenon of cellular footprint deposition": it is not clear whether the findings are indeed generic when investigating MDCK and Caco2 cell lines. Would the authors also observe an oscillatory cell movement, when using cell lines such as RPE1 cells, for which previously no oscillatory cell movement has been observed?

Could the authors pattern the 1D lines with different (in particular much higher) fibronectin concentrations and test whether they still observe an oscillatory movement? (See also major point 1)

Could the authors specifically inhibit fibronectin deposition (KO cell lines) to support their hypothesis that mainly the deposition of fibronectin drives the extension of migration boundaries?

Figure 1h: the reproducibility of the individual replicates is unclear: the graph is missing data points for the individual replicates.

Figure 1h: the text states that there is no difference in the frequency of the different cellular behaviours on different 1D track widths, yet the graphs show that the frequency is not entirely the same between e.g. 10 microns and 20/50 microns.

Lines 104/105: "significantly larger persistence time": Fig. 2b and 2c qualitatively show the cellular persistence, but its quantitative description (to calculate a significance) is unclear.

It will be important to control whether the oscillatory movements are still present without using mitomycin C.

We thank the reviewers for their comments which we address in details below. Their comments have been highlighted in blue and our answers are in black. In the revised manuscript, we have also highlighted in red the modifications that we made following these comments.

Reviewer 1

The manuscript by d'Alessandro, Barbier-Chebbah et al. reports on experiments and theoretical work done to investigate how cellular migration is affected by deposition of extracellular matrix. Using 2D micropatterns, it is demonstrated that cells leave footprints, in particular fibronectin, along their way, which determines future migration of the same cell on this pattern. Mathematically, the stochastic motion of cells is described as self-attracting random walks that exhibit anomalous subdiffusive behavior.

This manuscript addresses a fundamental question, namely, how extracellular matrix deposition can generate spatial memory that affects the migration patterns of cells. The study is well-designed and has been executed carefully. The combination of thoughtful experiments and a transparent mathematical model is appealing, both to biologists and physicists. Moreover, the manuscript is carefully written, very clear, and will therefore also be of interest for non-specialists. Therefore, the reviewer recommends publication of the work after an issue with the mathematical model has been fixed (see below) and some optional minor improvements have been made.

We thank the reviewer for this careful reading and positive comments on our manuscript.

The following points should be addressed:

-Figure 4a) The incremental msd curve from the PSATW simulations shows a maximum which is absent in the experimental data. This very pronounced difference between simulation and experiment should be explained and the model should be improved to fully capture the measurement results.

In the PSATW model as defined in the main text, in discrete time, indeed one observes a non monotonic behaviour of the mean squared displacement (MSD) – and even oscillations at short observation time T and delay time t . This non monotonic behaviours are due to the very low level of fluctuations in trajectories at short times (smaller than the cell persistence time). In particular, the first reversal events of cell polarity take place almost surely at the edge of the visited span, and occur at times that displays very small fluctuations. This induces quasi deterministic oscillations in the trajectories, which yield in turn non monotonic MSDs ; this behaviour disappears at larger times. To compare quantitatively this PSAW model to experimental cell trajectories, it is needed to effectively take into account both the fluctuations of cell velocities and cell/cell variability. To take into account fluctuations of cell velocities, we define in SI a continuous time version of the PSAW model, where the duration of each step is a random variable. To take into account cell-to-cell variability, for each trajectory generated numerically the parameter k (and thus $\beta = -2(k + 1)$) is drawn from a normal distribution of standard deviation $\simeq 0.5$ which reflects the variability observed in experiments. The effect of both fluctuations of cell velocities and cell/cell variability is shown in SI. Taking into account fluctuations of cell velocities and cell/cell variability, fluctuations in cell trajectories are enhanced at short times and the pseudo deterministic behaviour of the discrete PSATW model disappears. As a consequence, the MSD is monotonic, as observed in data. We show in SI that the parameters of the CTRW PSATW model, as well as the variability of k can be inferred from experimental data, and finally yields a good agreement between the predicted and observed MSDs. In the revised version of the manuscript, we modified the Figures 4a and d accordingly, and we added Supplementary figure 20 to discuss this effect.

-Title: The title “Cell migration driven...” is a bit misleading since the migration is driven by internal mechanisms while the authors discuss here extracellular guidance cues. One could instead write “Cell migration guided...”

The reviewer is right pointing that this word is misleading, and this remark goes in the same direction as another one from Reviewer 3. Therefore, we decided to replace “driven” by “guided” in the title as suggested.

-The authors mention that a subpopulation of the cells only spread while others undergo oscillatory motion. Can the authors elaborate on this finding? Why is this happening? What is the role of inhomogeneities in the printed contacts?

To our understanding, the main difference between “oscillatory” and “spreading” cells is that the former are able to polarise and move persistently – within the footprint span – while the latter remain unpolarised, or bipolar in this particular 1D geometry. Observing variability of cell polarity in cell assemblies is rather standard, and can be to some extent attributed to cell/cell variability. In the context of our experiments, one can expect that inhomogeneities in the substrate and the very interaction of cells with their footprint, which we demonstrate, could play a role in establishing – or not – cell polarity. These questions certainly warrant further research.

Line 243: a pronoun is missing in “...reciprocal dynamic adaptation between cell and its...”

We corrected this in the revised version of the manuscript.

Reviewer 2

1 Summary This manuscript presents results from a very thorough investigation of a subject that should have been addressed long ago by researchers studying cell motility on surfaces. The results presented combine rich experimental data with theoretical insights via a solid data analysis and computer simulations. This is science done very well. The high quality of this work and its presentation is obviously the result of a very close collaboration between experimenters and theorists—this is admirable. The manuscripts presentation is clear, in fairly perfect English. Conclusion: Publish after authors have considered my first item below and corrected the typos listed in the ensuing items.

We thank the reviewer for this careful reading and positive comments on our manuscript.

2 Details 1. Line 55: The references given here for oscillatory patterns in cell migration is nice. But they are from 2014 and 2019, and a much more striking observation of the phenomenon was published earlier, in 2011 in Ref. 13: Its Figures 4, 6, and 7 show distinct peaks/bumps in the power spectrum of velocities of individual, non-interacting amoebae, clear indicators of an oscillatory component (with drifting phase) to the self-propulsion of these single-cell organisms. Velocity auto-covariances shown in Ref. 13 show the same, thus confirming what cell trajectories shown in Ref. 13 seem to show: These single-cell organisms march to a rhythm of their own, though the phase of this rhythm drifts rather fast.

We thank the reviewer for bringing this to our attention. We added a reference to this work at this point in the manuscript.

2. Lines 198–199: “the fitted k values, that control the intrinsic cell persistence length, were...” ⇒ “the fitted k values, which control the intrinsic cell persistence length, were...” (if there is only one kind of k-values) OR “the fitted k values that control the intrinsic cell persistence length were...” (if there are more than one kind of k-values.)

3. Line 279: tough ⇒ though

4. Figure 4, caption: b. g. and d. h. “The long-time scaling exponent ... decreases...” ⇒ “The long-time scaling exponent, ..., decreases”

5. Line 289: See LaTeX manual for how to make correct quotation marks.

6. Reference 14: Bruckner ⇒ Brückner

7. Reference 19: Bruuckner ⇒ Brückner

8. Supplementary Fig. 2: Panels b and c of the figure have switched names in the caption.

We thank the reviewer for pointing out those typos that we corrected at the corresponding locations in the revised manuscript.

9. Supplementary Fig. 3: This figure is not mentioned/referred to anywhere in Supplementary Material. But the article does refer to it. Its caption should explain the strange-looking error bars of the figure.

This figure is referred to line 107-108 of the revised manuscript. We changed the representation of the data, using violin plots to show the full distribution of data points instead of our, more reduced, previous representation. We also added a panel **c** (quantification of the persistence of the movement) to answer a point raised by Reviewer 3.

Reviewer 3

Cells interact during locomotion with their microenvironment. During this interaction, cells do not only sense cues (chemical, mechanical, geometrical) in the extracellular matrix (ECM), but also remodel the ECM. Remodelling includes mechanical ECM rearrangement, ECM digestion (by e.g. secreting proteases), and ECM deposition (e.g. fibronectin). Deposition of ECM components by cells is well known to influence diverse biological aspects (e.g. Helvert, S. van, Storm, C. & Friedl, P. Mechanoreciprocity in cell migration. *Nature cell biology* 20, 8–20 (2018); Humphrey, J. D., Dufresne, E. R. & Schwartz, M. A. Mechanotransduction and extracellular matrix homeostasis. *Nature Reviews Molecular Cell Biology* (2014)).

d’Alessandro et al use patterned 1D lines - a well established method to analyse cell migration in a highly reductionist manner - and observe an unexpected oscillatory movement of MDCK (canine kidney) and Caco2 (colorectal cancer) cells. Monitoring this oscillatory movement for up to 96 hours revealed a continuously increasing length of the cellular path. By experimentation and theoretical modelling, the authors can convincingly show that this phenomenon can be explained by cellular deposition of ECM material (such as fibronectin) onto the oscillatory cellular paths and importantly onto the boundaries of the oscillatory cellular paths, thereby continuously increasing the length of the cellular path. Whereas the experiments are

largely convincing, the significance/relevance and some of the interpretations (cellular memory; space exploration) are questionable/unclear.

We thank the reviewer for this careful reading and judicious comments on our manuscript.

Major points:

Significance: several publications investigated migration on 1D lines. However, oscillatory cell movement on 1D lines has been so far only reported upon cellular manipulation, including zyxin-depletion (focal adhesion-/ actin stress fibre-protein; Fraley, S. I., Feng, Y., Giri, A., Longmore, G. D. & Wirtz, D. Dimensional and temporal controls of three-dimensional cell migration by zyxin and binding partners. *Nat Commun* 3, 719 (2012)) and microtubule-inhibition (Zhang, J., Guo, W.-H. & Wang, Y.-L. Microtubules stabilize cell polarity by localizing rear signals. *Proceedings of the National Academy of Sciences* (2014)), or upon usage of very defined ECM/pattern boundaries (Zhou, F. et al. Quasi-periodic migration of single cells on short microlanes. *PLoS ONE* 15, e0230679 (2020).; Brückner, D. B. et al. Stochastic nonlinear dynamics of confined cell migration in two-state systems. *Nature Physics* 15, 595–601 (2019)). In contrast, d’Alessandro et al observe oscillatory cell movements without cellular manipulation or defined pattern boundaries. This raises the question, whether the authors are looking at a very specific situation, in which the adhesiveness of the pattern (or of the particular cell lines) are just as adhesive to allow cellular binding to the pattern, but not adhesive enough to enable persistent cell migration without additional cellular ECM deposition.

The reviewer is accurate in this review on 1D migration. The fact that previous works did not evidence oscillatory migration in this geometry might be due to several factors. In particular, in order to observe those oscillations, we had to plate the cells at a very low density – to prevent footprints from distinct cells from overlapping – and acquire videos very soon after cell plating. These specific experimental conditions may not be the standard in previous publications. In addition, we believe that the data added following the reviewer’s suggestions (see the points below) demonstrate the robustness of our observations.

Interpretations: the title (“driven by long-lived spatial memory”) and the abstract (‘cells remember their paths’) could misleadingly imply that cells actively (and potentially intracellularly) generate a memory. Yet, the authors show in this manuscript that the extracellular deposition of ECM material (fibronectin) and subsequent binding to the ECM material influences the kinetics of cell migration. Their preconditioning experiment demonstrate that other cells than the depositing-cell can respond (as one would expect) to the deposited ECM - thus the findings are rather about deposition/detection cycles than about a memory.

The reviewer has it right ; the word “remember” can be misleading because it can take different meanings depending on contexts. Thus we changed it by “can retrieve their path” in the abstract. Yet, if one defines memory in a generic system merely as the fact that its dynamics at time t depends on the history of its trajectories at earlier times $t' < t$, as is often done in physics, the phenomenon that we describe is indeed a memory effect: be it isolated or in a group, the dynamics of a cell depends on the history of all cell trajectories. Interestingly, this class of memory effect does not require any complex information storage in the cell – and is therefore not “internal” – , but is merely induced by the interaction of the cell with its environment.

- Interpretations: the observed phenomenon rather confines cellular space exploration, as MDCK and Caco2 cells only increase their path length for short distances during every oscillation cycle. Thus, a title “Cell migration confined by..” instead of “Cell migration driven by..” could more accurately reflect the findings of the authors. Similarly, the authors should clarify this

aspect better in the abstract and discussion.

The reviewer is right pointing that “driven” might be misleading. Thus, and to also follow Reviewer 1’s suggestion, we changed the title for “Cell migration **guided** by...”.

Further aspects:

Line 123: “generic phenomenon of cellular footprint deposition”: it is no clear whether the findings are indeed generic when investigating MDCK and Caco2 cell lines.

The phenomenon is generic in so that it consists in a generic deposition-detection mechanism, which we highlight in the modelling part (for instance, it is retrieved in 2D as well). How frequently it is found depending on the cell types and environment is another – accordingly important – question, which certainly warrants further research (although our new data add some clues in that direction, see points below).

Would the authors also observe an oscillatory cell movement, when using cell lines such as RPE1 cells, for which previously no oscillatory cell movement has been observed?

Following the reviewer’s suggestion, we performed experiments with RPE1 cells on both control substrates and substrates conditioned by RPE1 cells. We observed that (*i*) on control substrates, even though some cells exhibit oscillations that look very similar to those of MDCK or Caco2 cells, a significant fraction of the cells are able to move persistently, and (*ii*) on conditioned substrates, RPE1 cells move over significantly larger distances than on control substrates. This is consistent with RPE1 cells being able to deposit a footprint along their path, with however a weaker interaction with the deposited signal. The repolarising signal at the edge of the footprint does not seem strong enough to fully confine all cells, even if at larger scales cell migration is still statistically slowed down. These data are displayed in the Supplementary Figure 12 of the revised manuscript.

Could the authors pattern the 1D lines with different (in particular much higher) fibronectin concentrations and test whether they still observe an oscillatory movement? (See also major point 1)

Following the reviewers recommendation, we sought to vary the protein concentration on the patterns and characterise the motion of MDCK cells on both such control and conditioned substrates. Thus, we varied the concentration of the fibronectin (Fn) solution during the first step of the process (adsorption on the PDMS stamps). The relative Fn concentration on the lines, measured by fluorescence intensity, was consistent with previous data of fibronectin adsorption on surfaces (see for instance Di Milla *et al.*, J. Cell Biol. vol. 122, 1993). In particular it showed that our usual condition is already close to the maximal possible concentration. In spite of these large variations in Fn concentration on the substrate, the cells behaved exactly similarly in all conditions. This suggests that the footprint does not simply consist in an increase of fibronectin concentration on the surface. Rather, the cells might deposit a mix of molecules (also see the point below), and arrange it in a way that is optimal for their migration, the fibronectin pattern being only used as a template. We display these data in the Supplementary Figure 14 of the revised manuscript.

Could the authors specifically inhibit fibronectin deposition (KO cell lines) to support their hypothesis that mainly the deposition of fibronectin drives the extension of migration boundaries?

We used siRNA to deplete fibronectin in Caco2 cells and then we assessed their migration

in our experimental set-up. Although the depletion was very efficient as shown by Western blot analysis ($\sim 95\%$), their motile behaviour on control and conditioned lines was undistinguishable from untreated Caco2 cells and cells transfected with a non-targetting siRNA. This might be explained by the fact that *(i)* fibronectin is also present in the serum and cells are able to assemble it on the surface, and *(ii)* the footprint may comprise other signals. Along this line, we showed by immuno-labelling that cells also deposit significant quantities of laminin. In the revised manuscript (lines 130–145), we highlighted that fibronectin alone might not be the only critical component of the footprint and we added two supplementary figures (Supp. Figs. 15 and 16) showing the effect of Fn KO and the deposition of fibronectin and laminin by both MDCK and Caco2 cells.

Figure 1h: the reproducibility of the individual replicates is unclear: the graph is missing data points for the individual replicates.

We modified Figure 1h to show the experimental variability.

Figure 1h: the text states that there is no difference in the frequency of the different cellular behaviours on different 1D track widths, yet the graphs shows that the frequency is not entirely the same between e.g. 10microns and 20/50 microns.

The sentences referring to this figure, stating that “both behaviours were approximately distributed” and then that “these observations were robust upon varying the width of the track” (p2 ll 51–61) arguably lead to think the frequency was the same on all track widths. We modified them to make this slight difference clearer.

Lines 104/105: “significantly larger persistence time”: Fig. 2b and 2c qualitatively show the cellular persistence, but its quantitative description (to calculate a significance) is unclear.

In order to quantify the persistence of a trajectory at a given time t , we measured the maximal distance it moved from the origin, and the cumulative length of its path. Then we computed the ratio of these two quantities: a value close to one reflects almost perfect persistence on a timescale t , while a value close to zero denotes a persistence time much lower than t . We added a panel **c** in Supplementary Figure 3 showing the distribution of this ratio at $t = 9 - 10$ h. While the distributions are peaked around 0.2 with a thin tail for control substrates, the distribution for conditioned substrates exhibits a secondary bump close to 1 denoting almost ballistic trajectories, while the primary bump peaked around 0.3 is also strongly skewed towards higher values.

It will be important to control whether the oscillatory movements are still present without using mitomycin C.

The reviewer is right pointing to this potential artefactual effect. We thus performed experiments with untreated MDCK cells on both control and conditioned substrates. Even though we cannot follow the cells for as long as with mitomycin C treatment, we clearly see oscillatory motion on control substrates and persistent motion on conditioned substrates. We added a mention to this experiment in the main text of the manuscript (p3, line 62) and a corresponding Supplementary Figure 13.

Reviewers' Comments:

Reviewer #1:

Remarks to the Author:

This work is of substantial interest for cell biologists and biophysics, which has already become clear from the first referee reports.

The manuscript has now been carefully revised to address the concerns that came up during the first round of review. All issues have been clarified. The authors also performed additional analysis and experiments to support their case.

Therefore, publication of the revised manuscript can be recommend.

Reviewer #3:

Remarks to the Author:

All major comments have been addressed.

There are only two minor aspects that should be addressed before publication:

Regarding the newly included data on RPE1 cells:

- the oscillatory movement of some RPE1 cells can be observed in Suppl Fig 12, yet the amplitude of oscillations does not seem to be changing over time, which is a central aspect in the manuscript and the other cell lines tested. The authors should comment on this difference in their manuscript.
- The authors state in the text that RPE cells on conditioned substrates "moved - slightly but significantly - further than cells on control substrates". The basis of this 'significance' statement is unclear, and visually not apparent in Suppl Fig. 12b.

The supplemental has a number of typos:

- e.g. Experimental procedures: "Prior to experiments, the cells were treated with 10 ug.mL added in the medium" - a word is missing here (what was added?)
- e.g. suppl. figure legend 16: e.g. "and lamini are visualised y immuno-staining"

Congratulations to the authors for their interesting findings and the elegant modelling.

Reviewer #1 (Remarks to the Author):

This work is of substantial interest for cell biologists and biophysics, which has already become clear from the first referee reports.

The manuscript has now been carefully revised to address the concerns that came up during the first round of review. All issues have been clarified. The authors also performed additional analysis and experiments to support their case.

Therefore, publication of the revised manuscript can be recommend.

⇒ We thank the reviewer for those supportive comments.

Reviewer #3 (Remarks to the Author):

All major comments have been addressed.

There are only two minor aspects that should be addressed before publication:

Regarding the newly included data on RPE1 cells:

- the oscillatory movement of some RPE1 cells can be observed in Suppl Fig 12, yet the amplitude of oscillations does not seem to be changing over time, which is a central aspect in the manuscript and the other cell lines tested. The authors should comment on this difference in their manuscript.

⇒ We added a comment on this difference page 5 (first paragraph) of the revised manuscript.

- The authors state in the text that RPE cells on conditioned substrates “moved - slightly but significantly - further than cells on control substrates”. The basis of this ‘significance’ statement is unclear, and visually not apparent in Suppl Fig. 12b.

⇒ This is true. We added a plot of the distance to origin after 16h in Supplementary Figure 12c. It shows data distribution which are clearly distinct although the difference (around 60%) between the two conditions is less than in other cell types (around 400% for MDCK and Caco2 in 1D).

The supplemental has a number of typos:

- e.g. Experimental procedures: “Prior to experiments, the cells were treated with 10 ug.mL added in the medium” - a word is missing here (what was added?)

- e.g. suppl. figure legend 16: e.g. “and lamini are visualised y immuno-staining”

⇒ We corrected those typos.

Congratulations to the authors for their interesting findings and the elegant modelling.

⇒ Finally we thank the reviewer for his challenging comments and fair assessment of our responses.